# A Novel Transcriptome Approach to the Investigation of the Molecular Pathology of Vitreous and Retinal Detachment

**DOI:** 10.3390/genes13101885

**Published:** 2022-10-18

**Authors:** Mel Maranian, Martin Snead

**Affiliations:** 1John van Geest Centre for Brain Repair, Cambridge CB2 0PY, UK; 2Department of Pathology, University of Cambridge, Cambridge, CB2 1TN, UK

**Keywords:** posterior vitreous detachment (PVD), retinal detachment (RD), rhegmatogenous retinal detachment (RRD), posterior hyaloid membrane (PHM), laminocyte, RNA sequencing (RNA-seq), spatial transcriptomics

## Abstract

Retinal detachment (RD) is one of the most common, sight-threatening ocular conditions requiring emergency intervention. Posterior vitreous detachment (PVD) occurs in the majority of an aging population whereby the vitreous body separates from the retina. It is well established that PVD is the common precursor to the most common forms of RD; however, it remains unknown why in most individuals PVD will cause no/few complications (physiological PVD) but in a small percentage will cause retinal tears and detachment (pathological PVD). Despite over 100 years of scientific research, the anatomical definitions of PVD and its pathogenesis remain controversial. Recent research has identified a novel cell population (laminocyte), present at significantly higher numbers in pathological PVD when compared to physiological PVD. We review and summarise the seven distinct clinical sub-groups of retinal breaks and focus on the role of the laminocyte in those secondary to PVD and the transcriptomic profile of this unique cell. Provisional whole transcriptome analysis using bulk RNA-Seq shows marked differentially expressed genes when comparing physiological PVD with PVD associated with RD. The limitations of bulk RNA-Seq are considered and the potential to address these using spatial transcriptomics are discussed. Understanding the pathogenesis of PVD-related retinal tears will provide a baseline for the development of novel therapeutic targets and prophylactic treatments.

## 1. Introduction

Retinal detachment (RD) is one of the most common, sight-threatening ocular conditions requiring emergency intervention with an estimated 7300 new reported cases each year in the UK alone. As of March 2020, there were 276,690 individuals registered as blind or partially sighted in the UK, with approximately 100 new sight impaired registrations due to rhegmatogenous retinal detachment (due to tears) reported annually [1,2]. In contrast to many other blinding retinal conditions, rhegmatogenous retinal detachment could be potentially avoidable or preventable if there was a clearer understanding of the cellular and genetic risk factors involved in its pathogenesis.

It is well established that posterior vitreous detachment (PVD), the separation of the vitreous body from the retina, can act as a precursor to several sight-threatening vitreoretinal pathologies including retinal tears and retinal detachment. PVD occurs in the majority of an aging population and is usually an uncomplicated benign ocular event. It remains unknown why PVD occurs in most but not all individuals and why it usually causes few or no complications, but in a small percentage causes retinal tears and detachment.

Conflicting anatomical definitions of PVD and its pathogenesis still prevail. Historically, the most prevalent concept of PVD is that it results from the separation of the dense outer collagen fibril layer of the vitreous, known as the vitreous cortex, from the internal limiting membrane (ILM) of the retina [3]. PVD is conventionally believed to be an inevitable, common, age-related event caused by structural changes of the vitreous humour, with liquefaction and eventual collapse separating it from the inner limiting membrane of the neurosensory retina. In youth, the innermost layer of the neurosensory retina and the vitreous demonstrate firm adherence to each other [4]. However, during the aging process, the vitreous degenerates during the combined processes of gradual liquefaction (synchysis) and degeneration and aggregation of collagen fibre arrangements (syneresis) [5,6]. One school of thought is that these combined biological changes cause contraction of the vitreous and the collagen fibres, weakening adhesion at the vitreoretinal interface, thus inducing it to pull away and separate from the retinal surface. However, this hypothesis does not fully account for the significant percentage of elderly individuals who do not undergo posterior vitreous detachment [7,8,9] or indeed the small number of young individuals with co-existing ocular pathologies that do.

More recent research has demonstrated that PVD is not simply an inevitable consequence of aging [10] but a more distinct immunohistochemically confirmed separation of the posterior hyaloid membrane (PHM) from the retinal surface [10,11,12]. In contrast to the main body of the vitreous, it has been shown that the detached PHM stains strongly for type IV collagen and therefore must form the inner aspect of the ILM prior to separation. These studies have also identified a novel cell population (laminocytes) integral to the PHM which are present at low density in physiological, uncomplicated PVD, but at much increased density in patients who have pathological PVD [12,13,14,15,16]. Structural deformation of the ILM associated with increased cellularity has been observed in patients with various vitreomaculopathies, including macular pucker and cellophane maculopathy [16]. This has led to the hypothesis that laminocyte activation may be a key molecular event in the development of a PVD and the vitreoretinopathies that ensue. The results of these studies now provide the foundation for new opportunities to investigate the cellular basis for the pathogenesis of PVD and retinal detachment.

## 2. Anatomy of the Human Eye

The human eye is a fluid-filled globe encompassed by three concentric layers of tissue: the sclera, uvea and retina (Figure 1). The sclera is a strong, optically opaque layer of fibrous connective tissue and is the outermost protective layer of the eyeball. Through rearrangement of collagen fibres, the sclera transforms at the front of the eye to the more specialised transparent tissue, the cornea, which allows light transmission into the eye through the pupil [17].

Immediately within the sclera lies the uveal tract comprising three continuous structures: (1) the iris, (coloured part of the eye), (2) the ciliary body, responsible for the suspension and adjustment of the refractive power of the lens, and the production of fluid (aqueous humour) between the cornea and iris, and (3) the choroid [17,18]. The choroid is an extremely vascular structure, with these vessels chiefly responsible for the nourishment of the outer retina [19] and choroidal abnormalities may result in neovascularisation and degenerative changes [20].

The third and innermost layer of the eye is the retina, comprising part of the central nervous system. It is the most sensitive part of the eye and is responsible for visual perception via conversion of light photons into neural impulses which are then transmitted via the optic nerve to the visual cortex [17]. The macula (macula lutea) is part of the central retina, located at the posterior pole of the eye. Although only accounting for < 4% of the retinal surface, it is responsible for almost all photopic vision, and RD that involves the macula will almost inevitably cause some degree of permanent central visual impairment [22,23].

Abutting the retina and filling the posterior cavity of the globe is the vitreous body, or humour; a colourless, transparent gel occupying approximately 80% of the posterior segment of the human eye with an average volume of 4 ml in the normal adult eye [24,25]. Vitreous humour is an extracellular matrix (ECM) composed almost entirely of water (~99%) [26] with hyalocytes, lipids, hyaluronic acid, inorganic salts and loosely packed collagen fibres accounting for the remaining volume [27,28].

## 3. Posterior Vitreous Detachment

Whilst the precise anatomical definition of PVD remains to be agreed, it is widely accepted that the separation of the vitreous from the retinal surface can precede sight-threatening conditions requiring medical intervention. More recent research into the pathogenesis of both physiological and pathological PVD [12,14,15,16] has identified and characterised the basement membrane (PHM) enveloping the detached vitreous, associated with a novel cell (laminocyte) population of varying proliferation. Since the PHM is only visible after PVD it must form part of the ILM of the retina in its attached state. Given that laminocytes are observed at a much higher density in pathological PVD when compared with physiological PVD, it is hypothesised that they are highly likely to play a role in PVD and the pathogenesis of various secondary vitreoretinal disorders.

## 4. Retinal Detachment

Retinal detachment (RD) is one of the most common ocular conditions requiring emergency intervention, with an estimated 7300 new cases each year in the UK alone [29] and is the most common potential cause of blindness following cataract surgery [30]. Retinal detachment results from the separation of the neurosensory retina from the underlying retinal pigment epithelium (RPE) [31]. It is considered as one of the few true ocular emergencies and without medical intervention, can cause permanent severe vision loss or blindness. There are three main types of retinal detachment; rhegmatogenous, tractional, and exudative, all of which demonstrate significant differences in terms of their frequency and pathophysiology.

## 5. Rhegmatogenous Retinal Detachment

Rhegmatogenous retinal detachment (RRD) refers to retinal detachment secondary to a retinal tear(s) and is the most prevalent type of retinal detachment. The vast majority result from retinal breaks or tears that often occur as a result of PVD. RRD has a reported incidence of 6.3–17.9 per 100,000 population [32]. RRD results from the separation of the neurosensory retina (NSR) from the retinal pigment epithelium (RPE) underneath enabling retrohyaloid, preretinal fluid to pass through the tear/s and accumulate in the sub-retinal space [33]. Multiple factors including myopia, ocular trauma, inherited vitreoretinopathies, lattice degeneration, and cataract surgery as well as racial differences are known to be associated with the risk of RRD development [34] but the precise pathways involved in its pathogenesis are complex and not yet fully understood [35]. Ethnicity is strongly associated with the incidence of RRD; age adjusted rates have been reported as 10.5 per 100,000 in Singapore, 11.6 per 100,000 in Chinese, 7.0 per 100,000 in Malay and 3.9 per 100,000 in Indian populations [36]. Incidence in black South Africans is considerably lower at 0.46 per 100 000, the reasons for which remain unknown [37]. Such variability clearly demonstrates the evidence of a genetic component to the risk of RRD, also supported by familial aggregation studies showing a significantly higher incidence of RRD in siblings and offspring of cases, independent of sex, age, or myopia [38].

Deducing the comprehensive genetic architecture of RRD is complex. Several studies have established heritable factors in its pathogenesis, of which the Stickler syndromes are perhaps the most obvious example. Other genetic risk factors identified to date require further investigation [39]. In the case of disorders such as retinal detachment, association studies are severely disadvantaged by the difficulty of recruitment of sufficient subject numbers. Attempts to bolster numbers of affected individuals risks degrading any observable association by amalgamating different phenotypic sub-groups of retinal detachment, which potentially have different underlying molecular mechanisms [39].

Within RRD, there are at least seven well recognized distinct clinical sub-groups of retinal breaks, some associated with PVD and other less common sub-types in which the vitreous gel remains attached [40]. It is likely that each sub-type will have a unique genetic signature and that more stringent classifications/stratifications are required in future cohorts. This may prove problematic given the limited sample sizes of most studies, with any GWAS requiring large (>10,000) sample sizes, for most disorders, in order to yield valid, reproducible results [41]. This review focuses solely on those sub-types secondary to PVD (Table 1) [40].

### 5.1. Horseshoe Retinal Tears

Horseshoe tears (HST) often appear as U-shaped retinal “flaps”, due to vitreous detachment, causing full-thickness breaks of the neurosensory retina (Figure 2). The estimated incidence of a horseshoe retinal tear in patients with acute PVD is 8%, with this type of retinal tear accounting for over 80% of all cases of retinal detachment [51,52]. This sub-group is also the most urgent with rapid progression to macular involvement.

The 1202 cases of RRD reported by Mitry et al. were included as part of the largest known genome wide association study (GWAS) of RRD to date [52]. Eleven significantly associated variants were identified, lying within or proximal to, *LOXL1*, *DLG5*, *EFEMP2*, *TRIM29*, *COL2A1*, ***ZC3H11B PLCE1***, ***FAT3***, ***BMP3***, ***TYR***, and ***COL22A1***. Of these 11 loci, six (indicated in bold) were replicated in the 23andMe data set, where RD was self-reported by participating individuals [53].

### 5.2. Giant Retinal Tears

Giant retinal tears (GRTs) may be defined as full thickness retinal tears occurring at the pars planar junction with a circumference of > 90°, in the presence of a PVD [54,55,56]. Large population-based studies have shown this condition to be more common in males (71.7%) with a mean age of 42 years [57]. It is estimated to comprise 1.5% of all reported rhegmatogenous retinal detachments with a risk of bilateral GRT (involvement of the fellow eye) of 12.8% but with a much higher incidence of bilaterality in various hereditary vitreoretinopathies, especially Sticker syndrome. As the vitreous separates as far anteriorly as the pars plana, GRTs commonly progress rapidly to extensive retinal detachment, unlike cases of retinal dialysis which may also extend beyond 90 degrees but where the attached vitreous remains attached so that independent mobility of the posterior flap is not a feature [58]. Aside from Stickler syndrome (see Soh et al. and Alexander et al., this issue), other risk factors for GRTs include high myopia, aphakia, pseudophakia [59,60] and both penetrating and blunt ocular trauma, of which one study found the latter to be associated with 25% of all cases [61].

### 5.3. Macular Hole-Associated Retinal Detachment

Macular hole-associated rhegmatogenous retinal detachments are commonly seen in patients with highly myopic eyes and posterior staphyloma [62]. A posterior staphyloma has been described as an outpouching of the ocular wall and is a typical hallmark in pathological myopia [63,64]. Although uncommon, this form of RRD is also associated with PVD.

## 6. The Laminocyte

Twenty years have passed since Snead et al. identified a novel cell population, integral to the PHM, for which they proposed the term ‘laminocyte’. This name was chosen to highlight both its basement membrane association (of which type IV collagen and laminins are essential structural components) and additionally the laminar pattern of their array entirely within the plane of the detached PHM but absent from the main body of the vitreous gel itself [13,65]. Decades of earlier publications reported the presence of vitreal cells associated with the vitreoretinal interface but failed to classify them morphologically or via immunohistochemistry [66,67]. The laminocyte has both a different immunohistochemical profile and distribution from the resident hyalocyte found within the cortical gel [68]. Hyalocytes exhibit features suggestive of macrophage lineage [69], although lack expression of the common macrophage marker CD68 and GFAP (glial fibrillary acidic protein) an intermediate filament (IF) protein highly specific for cells of astroglial lineage [27,70,71,72]. Hyalocytes are also known to express S100 protein, known to exert regulatory effects on a broad range of cell types, including macrophages [73,74,75]. Conversely, the laminocyte demonstrates positive staining for both CD68 and GFAP, and negative staining for S100 protein providing convincing evidence that the laminocyte is immunohistochemically distinct from the hyalocyte [68].

Light microscopy studies of the histological and immunohistochemical profile of the PHM and associated resident laminocytes from post-mortem donor globes reveal a spindle like shaped morphology, with an indistinct cytoplasm and round or oval nuclei (occasionally appearing binucleated) and a branched, extensive dendritic morphology. Laminocytes appear most densely populated at the Weiss ring (a circular attachment of glial tissue observed in the vitreous body after detachment from the optic nerve head), becoming increasingly sparsely populated in the anterior PHM [68].

Immunohistochemical studies demonstrate positive staining of the PHM, for both laminin and collagen IV, (Figure 3a) providing additional evidence that this structure is a distinct basement membrane with the laminocytes integrally associated on scanning electron microscopy (Figure 3b). Figure 3c shows a single laminocyte captured by confocal microscopy [68].

## 7. Discussion and Future Projections

The last 50 yrs have seen numerous advances in the microsurgical techniques for retinal detachment repair but the pathogenesis of retinal detachment and its precursor PVD remains poorly understood, particularly the cellular basis that differentiates pathological from physiological PVD. Research is currently in progress utilising a transcriptome-based RNA-Seq approach to explore differential gene expression in PHM samples from patients with either physiological uncomplicated PVD or PVD associated with RD.

Since RRD is known to comprise a heterogeneous group of at least seven distinct sub-types, with likely differing aetiologies, these current investigations are focussed on retinal tears secondary to posterior vitreous detachment to investigate the transcriptional changes associated with cellular proliferation of the laminocyte in pathological PVD [12,14,15,16].

The term transcriptome denotes the entirety of all RNA transcripts present within a cell at a given time and its analysis provides valuable insight into the functional elements of the genome. In the late 1990′s and 2000′s, the progression of DNA arrays (or microarrays) was rapid and was the dominating method employed in large studies of gene expression [76,77]. Microarrays utilize the immobilisation of thousands of known nucleic acid fragments on to a solid surface (such as a glass slide), to which complementary sequences of sample material can hybridise and be measured via chemical fluorescence.

RNA sequencing (RNA-Seq) utilizes high-throughput next generation sequencing (NGS) technology to help decipher the complex architecture of the transcriptome of a cell and whilst more costly, in the last few years, its use is proving superior to the use of microarrays [77,78]. RNA-Seq simultaneously measures the expression of thousands of genes and is not subject to the common limitations of hybridisation techniques, including the requirement of a priori knowledge of existing sequences, probe performance, and potential cross hybridisation from homologous transcripts [77]. RNA-Seq also detects and measures novel transcripts, gene fusions, splice variants, point mutations and single nucleotide polymorphisms (SNPs) and generates a more informative genetic profile than microarrays [79]. There are multiple types of RNA, of which the most commonly studied species is mRNA, specifically encoding proteins [80]. However, whole-transcriptome analysis using total RNA sequencing permits the detection of both coding and various classes of non-coding forms of RNA (involved in multiple cellular responses and functions) [81] and therefore yields the most comprehensive view of the transcriptome.

Current research involves the collection of vitrectomy samples from patients undergoing surgical repair of rhegmatogenous retinal detachment secondary to PVD and comparison with samples from individuals with physiological PVD. Briefly, vitrectomy washings are filtered using cellulose nitrate membrane vacuum filter units. TRIzol^®^ reagent was added and a cell scraper used to agitate and retrieve any cellular material present on the filter membrane. Samples are stored in Eppendorfs at −80 °C prior to RNA extraction. Following RNA extraction, next generation sequencing (NGS) libraries are constructed from all samples yielding sufficient material and sequenced on an Illumina platform, allowing analysis of the vitrectomy and PHM whole transcriptome.

Whilst this review focusses on samples obtained from cases of rhegmatogenous retinal detachment, vitreous samples are also being investigated from a variety of other vitreoretinal disorders including macular hole and cellophane maculopathy procedures (from patients both with and without associated PVD). Samples from individuals without PVD collected as controls yield no RNA, in contrast to successful RNA extractions from patients with PVD. This lends further supportive evidence that the transcriptome profiles obtained include the distinct cellular signature associated with the detached PHM and are not the result of blood or other tissue contamination from the surgical entry sites.

Provisional data analysis shows markedly different expression profiles when comparing physiological (uncomplicated) PVD against PVD associated with RRD, with the overwhelming majority of upregulated genes associated with the pathological group (Figure 4).

Using such a bulk RNA-Seq approach averages global gene expression of all cells in any sample obtained with the potential to mask the transcriptome profile of the novel laminocyte by other cell populations collected during surgery, including those from other membranes or blood contamination. The negative control samples strongly suggest that such contamination is highly unlikely but further analysis is required to isolate and more accurately define the unique profile of the laminocyte. Furthermore, sample analysis is both complex and challenging given the microscopic size of tissue retrieved at surgeries.

To address this difficulty, parallel investigations utilising novel GeoMx digital spatial transcriptomic (DSP) technology (NanoString, Seattle, WA, USA) are also in progress. This innovative methodology has been optimised for detection of RNA and/or protein from whole-tissue samples, including valuable formalin fixed paraffin embedded (FFPE) sections. This permits assessment of the whole transcriptome of multi and near-single cell populations (specifically in this case, the laminocyte) in situ, retaining the composition and spatial organization within the PHM from patients with both physiological PVD and RRD for comparison. This serves as a powerful tool to explore tissue pathology, architecture and gene expression profiles in both healthy and abnormal specimens. Briefly, tissue sections are mounted on to glass slides to which over 18,000 hybridisation probes targeting human transcripts are added. A unique molecular identifier (UMI) is incorporated into each UV-cleavable probe. With the use of preselected morphological markers, users can select specific regions of interest (ROIs) and subsequently cleave and collect the probes in any chosen region. Using NGS, UMIs are counted from each ROI providing transcriptomic data with spatial and morphological context [82]. Given the paucity of intraocular tissue available for investigation, this technology has the added advantage of access to the considerable historical resources of post-mortem FFPE samples. Fresh tissue available from routine surgical procedures and spatial transcriptomic analysis of both fresh and fixed tissue in parallel would also serve to disclose any discrepant results between sample age and/or fixation and preservation methods.

It is hoped that the identification of differentially expressed genes and their biological functional pathway in laminocytes associated with PVD will shed new light on how and why this is a relatively benign event in the majority of individuals but leads to retinal tears and retinal detachment in the minority. If the pathogenesis was elucidated more clearly, the risk of PVD-related retinal tears could be more accurately stratified, and thus provide a baseline for the development of novel therapeutic targets and prophylactic treatments.

## Figures and Tables

**Figure 1 genes-13-01885-f001:**
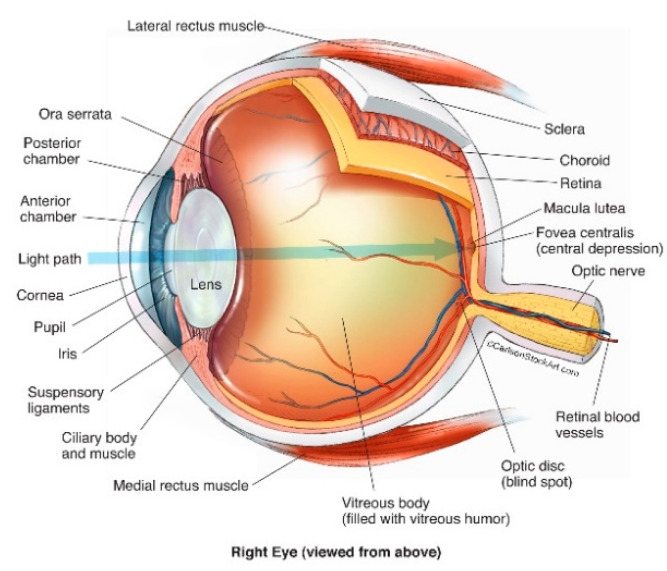
Structure of the human eye (© Reprinted/adapted with permission from Ref. [21] (accessed on 25 May 2022).

**Figure 2 genes-13-01885-f002:**
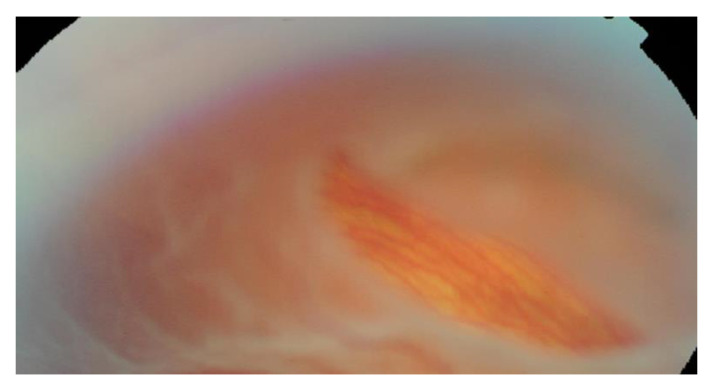
Horseshoe retinal tear.

**Figure 3 genes-13-01885-f003:**
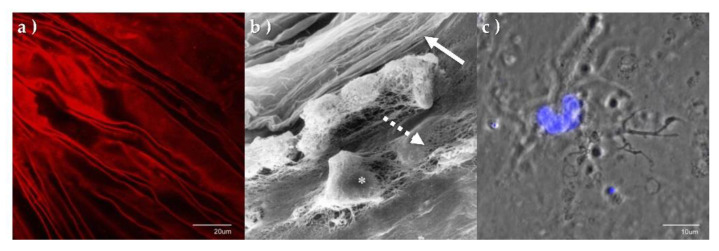
Immunohistochemistry and electron microscopy of PHM and laminocytes. (**a**) PHM confocal surface topography micrograph stained with collagen IV antibodies, demonstrating distinct creased appearance. (**b**) High power scanning electron micrograph of the PHM and associated laminocytes. (* laminocyte). The distinct folded vitreal aspect of the PHM (solid arrow) can be seen as a separate structure enveloping the fibrillary gel of the vitreous body (dashed arrow). (**c**) Confocal micrograph of single laminocyte reveals unique ‘large ‘cashew’ shaped nucleus. (Tissue stained with DAPI, x100.8 magnification).

**Figure 4 genes-13-01885-f004:**
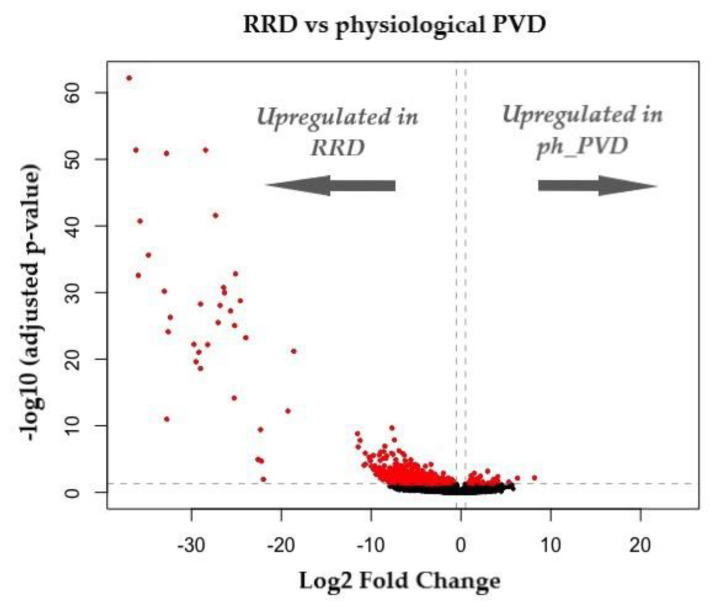
Volcano plot representing DEGs (differentially expressed genes) between RRD and physiological PVD. Log2 FC indicates the mean expression level for each gene. Each dot represents one gene. Red dots are indicative of statistically significant genes (adjusted *p* < 0.05) and log2FC < 0.5) (unpublished data).

**Table 1 genes-13-01885-t001:** The ‘Cambridge Guide’ to the features associated with the seven most common types of primary retinal break in rhegmatogenous retinal detachment. Reproduced and adapted with permission from Ref. [40]. This review focusses on those sub-groups secondary to PVD (highlighted in bold).

Break Type	PHM * Status	Vitreous Architecture	Sex	Typical Age Group (Years)	Refractive Error	Fellow-Eye Involvement Pathology	Reference
Atraumatic dialysis	On	Normal	M > F	8–20	Emmetropia/hypermetropia	5–15%	[42]
**Giant retinal tear**	**Off**	**Check for anomaly**	**M = F**	**5–50**	**Moderate/** **high myopia**	**Variable up to 80%**	[43,44]
**Horseshoe tear**	**Off**	**Usually** **syneretic**	**M = F**	**45–65**	**Moderate/** **high myopia**	**10%**	[45]
Round retinal hole	On	Usually normal	F > M	20–40	Moderate myopia	45%	[40]
**Macular hole**	**Off**	**Syneretic**	**M = F**	**45–65**	**High myopia**	**Rare**	[46,47]
Degenerative retinoschisis	On	Normal	M > F	70+	Hypermetropia	80%	[48,49]
X-linked retinoschisis	On	Normal, may have haemorrhage	M	10–20	Emmetropia	100%	[50]

* PHM = posterior hyaloid membrane; Macular hole refers to macular hole associated with retinal detachment, distinct from isolated ‘idiopathic’ macular hole.

## Data Availability

Not applicable.

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
