# Peer review of "A Novel Transcriptome Approach to the Investigation of the Molecular Pathology of Vitreous and Retinal Detachment"

_genes, 2022, doi:10.3390/genes13101885_

Round 1

Reviewer 1 Report

The contribution by Maranian and Snead leaves you with great expectations after reading the title. It is a well written peace. RD is certainly of great concern and deserves intense studies to understand all details about it.

Nevertheless, I have some serious concerns.

1.     I am not sure what the rational is. Are the authors trying to follow the question of the cause of physiological PVD? Why? How do they collect the appropriate material? Answers to these questions are not clearly presented.

2.     I am not sure whether this is a research article or a review article or a research proposal. It seems a hybrid of all of the above. 

3.     It is somewhat disturbing to find incomplete references. There are good programs that allow downloading complete references. Please check Ref 4.

Reviewer 2 Report

In this review article, the author discussed posterior vitreous detachment (PVD) and its related retinal detachment.  Detailed information including human eye anatomy, PVD, and retinal detachment is provided.  Various types of rhegmatogenous retinal detachment are described, which is frequently associated with PVD.  Furthermore, new cell population, the laminocyte, that localize at posterior hyaloid membrane (PHM), has been described.  Genetics and GWAS results of RRD are also discussed.  Finally, the author consider transcriptomics study of laminocyte might provide new mechanistic insights of PVD.  Overall the review is well written and informative.  My comments are below:

1.     The authors propose to focus on laminocyte.  What is known liteature supporting relationship between laminocyte and PVD?

2.     Is the change of density of laminocyte a result of PVD or cause of PVD?

3.     GeoMx is not true single cell resolution.  What is the rationale of choose GeoMx vs other technologies, such as SeqFish and MERFISH, etc?

4.     For the DEGs observed, any overlap with GWAS hits?

Round 2

Reviewer 1 Report

I am glad to read that this is a review. Would be nice to read that at the very beginning.

Thanks for adding the method part (line 288-303). I was not necessarily after this, although it is informative. I am rather interested in how the samples were identified and collected from patients with physiological PVD, rather than pathological PVD. This is still missing. Also, it would make a more convincing concept if the number of patients/samples would be given. Further, it is curious that no RNA could be isolated from control persons; this might be worth a comment or reasoning.

Thanks for correcting the reference, which is also in the authors interest.
